# Optimizing the design of spatial genomic studies

Andrew Jones[1,6], Diana Cai [2,6], Didong Li[3] & Barbara E. Engelhardt [4,5] ✉

Spatial genomic technologies characterize the relationship between the structural organization of cells and their cellular state. Despite the availability of various spatial transcriptomic and proteomic profiling platforms, these experiments remain costly and labor-intensive. Traditionally, tissue slicing for spatial sequencing involves parallel axis-aligned sections, often yielding redundant or correlated information. We propose *structured batch experimental design*, a method that improves the cost efficiency of spatial genomics experiments by profiling tissue slices that are maximally informative, while recognizing the destructive nature of the process. Applied to two spatial genomics studies—one to construct a spatially-resolved genomic atlas of a tissue and another to localize a region of interest in a tissue, such as a tumor—our approach collects more informative samples using fewer slices compared to traditional slicing strategies. This methodology offers a foundation for developing robust and cost-efficient design strategies, allowing spatial genomics studies to be deployed by smaller, resource-constrained labs.

Spatially-resolved genomic assays present an opportunity to study the physical organization of cells and how cell phenotypes vary across space[1]. These assays have been used to study a variety of biological tissues and organs, such as brain[2,3], liver[4], heart[5], and various tumors[6]. However, spatial genomics experiments are costly in terms of financial, labor, and material resources. Ideally, an experiment studying a particular tissue type would collect genomic data—such as gene expression or protein expression—from the entire spatial domain of the tissue of interest. However, cost-constrained scientists are typically forced to select only a small fraction of a 2D slice of the tissue to profile—the field of view (FoV).

A common approach to collecting a spatial genomic sample is to identify an anatomical region within the tissue of interest and take one or multiple parallel cross-section slices of the tissue in the region of interest. This approach is attractive for its simplicity in collecting the slice and for its adherence to orthodox slicing strategies (i.e., slicing along primary anatomical axes, such as coronal or sagittal axes). Despite its attractive properties, this data collection approach may not provide the most informative data. In particular, adjacent, parallel

cross-sections may contain largely redundant information as opposed to more distant, potentially non-parallel slices. Furthermore, taking a slice from a tissue is inherently destructive, splitting the tissue into two pieces to prohibit future slices from intersecting any prior slice. However, because tissues are sliced when the tissue is frozen, iterative experiments are possible because the tissue is never thawed. There is a need for a systematic approach to designing spatial genomics experiments—in particular, optimizing the choice of which tissue cross-sections to collect—such that the experiments are maximally informative for the task given the previous slices.

In this paper, we propose a statistical approach to optimize experimental design for spatial genomics studies. Specifically, we focus on the problem of choosing which cross-sections of a tissue will yield a maximally-informative experiment for a spatial genomics assay. Our proposed approach relies on fundamental concepts in Bayesian optimal experimental design (BOED)[7]. For a given statistical model of the data, our method finds the slice that is expected to provide the maximum amount of additional information about the tissue of interest given the prior information contained within the earlier slices,

---

[1]Department of Computer Science, Princeton University, Princeton, USA. [2]Center for Computational Mathematics, Flatiron Institute, New York, USA. [3]Department of Biostatistics, University of North Carolina at Chapel Hill, Chapel Hill, USA. [4]Gladstone Institutes, San Francisco, USA. [5]Department of Biomedical Data Science, Stanford University, Stanford, USA. [6]These authors contributed equally: Andrew Jones, Diana Cai. ✉e-mail: bengelhardt@stanford.edu

and also considering the current fragmentation of the tissue from removing the earlier slices. Our framework allows for designing experiments with a variety of experimental goals and can be adapted to the experimenter's preferred statistical modeling approach. We demonstrate our approach through two different experimental goals: building a 3D tissue atlas and localizing a tumor within a tissue.

## Related work

**Optimal experimental design**. The literature on optimizing and automating experimental design has a long history[8–11].

Optimal design first arose in the frequentist literature, and specifically in the setting of linear models (see[12] for an early review). Frequentist approaches to experimental design start by positing an optimality criterion. Let $\mathbf{X}$ be an $n \times p$ design matrix where the experimenter is tasked with choosing $n$ designs from a design space $\mathcal{X} \subset \mathbb{R}^p$; we define the *information matrix* as $\mathbf{X}^\top\mathbf{X}$. A design $\mathbf{X}^\star$ is optimal with respect to a criterion, denoted by $\mathcal{L} : \mathcal{X} \to \mathbb{R}$, if it satisfies $\mathbf{X}^\star = \arg\max_{\mathbf{X} \in \mathcal{X}} \mathcal{L}(\mathbf{X})$. Several criteria have been proposed, such as *D*-optimality, where $\mathcal{L}(\mathbf{X}) = \det(\mathbf{X}^\top\mathbf{X})$; *A*-optimality, where $\mathcal{L}(\mathbf{X}) = -\operatorname{tr}((\mathbf{X}^\top\mathbf{X})^{-1})$; and *E*-optimality, which maximizes the minimal eigenvalue of $\mathbf{X}^\top\mathbf{X}$.

Bayesian optimal experimental design (BOED)[7] extends experimental design to the Bayesian setting, where the model parameter $\theta$ is assumed to be random and drawn from a prior distribution $\pi(\theta)$, and each observation $y$ is drawn from a likelihood model $p(y|\theta, x)$. Given a vector of $n$ observations $\mathbf{y} \in \mathbb{R}^n$, standard Bayesian inference proceeds by computing the posterior distribution, $p(\theta|\mathbf{y})$. Similar to the frequentist approach to experimental design, BOED techniques seek designs $\mathbf{X}$ that are expected to improve statistical inference in some way. However, unlike the frequentist setting that optimizes a function of a frequentist estimator, BOED approaches seek to improve the posterior according to some criterion. One of the most popular criteria is the expected information gain (EIG), which is defined as the expected difference in entropy between the prior and posterior:

$$\mathrm{EIG}(\mathbf{X}) = \mathbb{E}_{\mathbf{y}}\left[H[p(\theta)] - H[p(\theta|\mathbf{X}, \mathbf{y})]\right], \tag{1}$$

where $H[p(\omega)] = -\int_\Omega p(\omega)\log p(\omega)d\omega$ is the differential entropy of a density function $p(\omega)$. A design that maximizes the EIG is also called Bayesian *D*-optimal because, in the case of a linear model, maximizing the EIG is equivalent to finding a *D*-optimal design. The EIG may also be written in terms of predictive distributions in place of posterior distributions (see Supplementary A.1.2 for a derivation):

$$\mathrm{EIG}(\mathbf{X}) = \mathbb{E}_{\theta}\left[H[p(\mathbf{y}|\mathbf{X})] - H[p(\mathbf{y}|\theta, \mathbf{X})]\right]. \tag{2}$$

Depending on the setting, one form of the EIG may be easier to work with than the other. For complex statistical models, computing the posterior and predictive distributions analytically is impossible, making the EIG computationally intractable as well. Recently, approximate inference methods for BOED have been proposed that ease the computational burden[13–15]. However, these approaches come at the expense of an exact solution.

**Experimental design for genomics studies**. In genomics, automating experimental design has become of special interest in recent years due to the rising cost and complexity of experimental protocols. Several statistical approaches have been proposed for designing single-cell sequencing experiments[16–19].

In the field of spatial genomics, a technique was recently developed for determining the appropriate experimental parameters to achieve a desirable level of statistical power[20]. Despite these advances, there remains a lack of methods to optimize the physical locations of a tissue to profile.

## Results

### Experimental setup

We now demonstrate our experimental design approach through application to synthetic data and three spatial gene expression datasets. Throughout our experiments, we compare five methods for experimental design:

- *EIG*: Maximize EIG over candidate cross-sections while accounting for tissue fragmenting.
- *EIG (parallel)*: Maximize EIG over candidate cross-sections while constraining the slices to be parallel and along an anatomical axis.
- *EIG (no fragmenting)*: Maximize EIG over candidate cross-sections, allowing for slices to cut across multiple tissue fragments.
- *Serial*: Take serial parallel cross-sections along an anatomical plane.
- *Random*: Randomly choose from candidate cross-sections while accounting for tissue fragmentation.

The first three approaches, *EIG*, *EIG (parallel)*, and *EIG (no fragmenting)* are special cases of our proposed approach with different design spaces. The *Serial* design approach is the one most commonly used in spatial genomics experiments. The *Random* approach would never be used in practice but serves as a baseline comparison.

In the following experiments, we consider cross-sections through a tissue, where the tissue is represented by a cloud of points. As described above, each cross-section is defined by a plane. We define a slice's width as $\delta$ and allow any spatial location within distance $\delta$ to the plane to be observed after taking this slice. In practice, the choice of the section width $\delta$ is often dictated by the data collection modality being used, as well as the tools available.

### Simulations

We first conducted a series of simulation studies in order to evaluate the behavior of our approach to structured batch experimental design.

**Small-scale demonstration**. As an initial demonstration and visualization of our approach, we applied our method for atlas building to a simulated tissue. For ease of visualization, we first study a setting where we are tasked with choosing one-dimensional slices (lines) from a two-dimensional tissue. The simulated tissue had a circular shape with a radius of five (Fig. 1a). We placed points randomly within the boundaries of the tissue, which represent the locations of cells or spots. We then generated synthetic responses at each location using a GP with a mean of zero and Matérn 1/2 covariance function with length scale $\ell = 1$ and noise variance $\tau^2 = 0.1$.

On each iteration of this experiment, the objective is to choose a one-dimensional cross-section through the synthetic tissue. After taking a slice, the observations from cells lying within distance $\delta$ of the slice's line, along with the response value at each of these cells, are revealed (Fig. S1a, b). The tissue then splits, going from $t$ to $t + 1$ fragments (Fig. 1b–d). We then repeat this process for a total of $T$ iterations.

We applied our experimental design approach to find the EIG-optimal experimental design on each iteration. To do so, we first discretized the space of possible designs. We parameterized each design by its angle $\phi$ with the $x$-axis and its intercept $b_0$ with the $y$-axis, and took 50 slopes whose angles are equally spaced in $[0, \pi)$ and 50 intercepts equally spaced in $[-5, 5]$. (The slope of the line is given by $\tan\phi$.) Using all pairwise combinations of $\phi$ and $b_0$, this resulted in a design space containing 2500 cross-sections.

We ran our method forward for $T = 10$ iterations. On each iteration, we computed the EIG for each possible slice and selected the slice with the highest EIG (Fig. 1e). We then visualized the resulting slices. In general, the optimal slices under our criterion tended to be the cross-sections that intersected the most cells (Fig. 1g; Fig. S1c). On the first iteration, a slice near the center of the circular domain was chosen. On subsequent iterations, slices that were somewhat parallel to the first

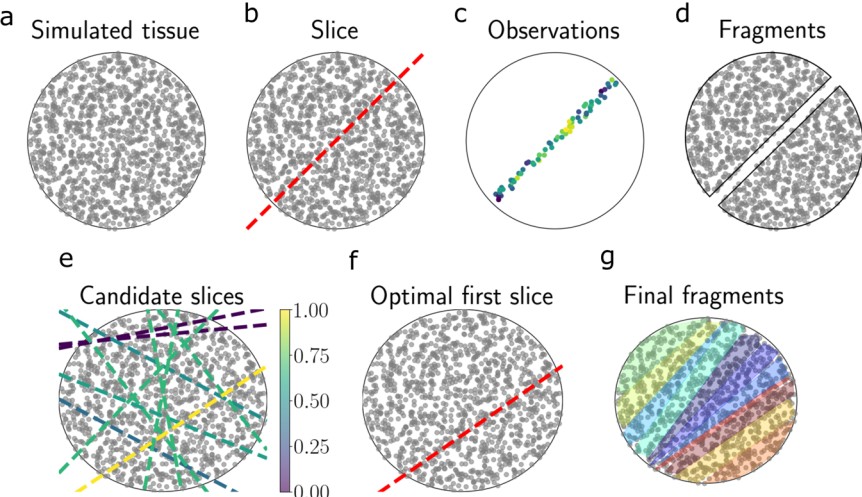

**Fig. 1 | Demonstration of slicing in two-dimensional simulated tissue.**
**a** Simulated spherical tissue with a grid of spots. **b** An example one-dimensional slice through the tissue. **c** The resulting observations at each spot after taking the slice in (**b**). The colors represent a univariate phenotype. **d** After slicing, the simulated tissue is split into two fragments. **e** Each line represents a candidate one-dimensional slice. Each slice is colored by its EIG (normalized to have a maximum of one). The slice with the highest EIG is then chosen; see panel (**f**). **f** The EIG-maximizing slice from the candidates in panel (**e**). **g** Tissue fragments after $T = 10$ iterations of repeated slicing. Each color represents a distinct fragment. Source data are provided as a Source Data file.

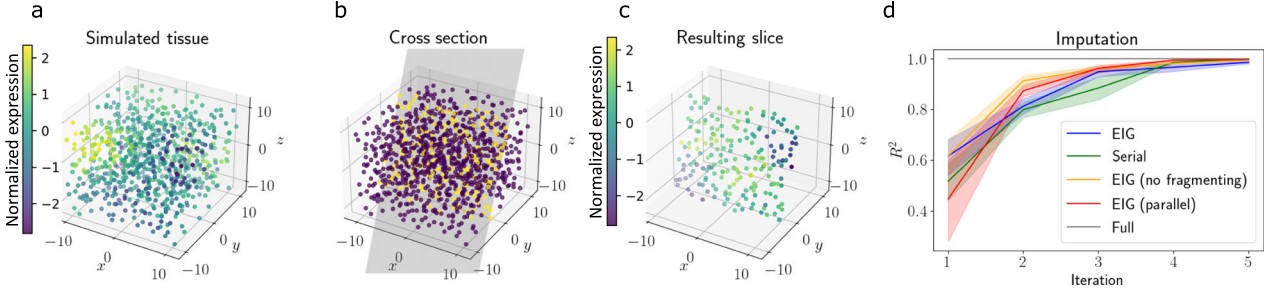

**Fig. 2 | Imputing unobserved gene expression from observed cross-sections.**
**a** Simulated tissue colored by synthetic gene expression. **b** An example slice through the synthetic tissue. **c** The resulting observations from the slice in (**b**). **d** $R^2$ for gene expression imputation after each slicing iteration for each method. Error bands represent 95% confidence intervals computed using $n = 5$ runs. Source data are provided as a Source Data file.

slice were chosen. We repeated this experiment with non-uniformly spaced cells and found a similar improvement in the design (Fig. S2). This demonstration suggests that EIG maximization under the atlas model encourages choosing slices including many cells over slices containing few cells.

**Three-dimensional demonstration.** We next conducted a similar experiment, but this time we extended it to a three-dimensional spatial domain. We placed points randomly within a cube with edge length 10 and generated synthetic responses at each location from a GP with a radial basis function (RBF) covariance function with length scale $\ell = 1$ and noise variance $\tau^2 = 0.1$ (Fig. 2a). We ran our experimental design procedure for $T = 10$ iterations. To quantitatively evaluate the chosen cross-sections, we ran a prediction experiment on each iteration. Specifically, after iteration $t$, we fit a GP with an RBF covariance function using the data collected theretofore, estimating the RBF hyperparameters using maximum likelihood estimation. We then computed the predictive mean for the unobserved spots and computed the goodness-of-fit $R^2$ between the predictions and the true values. Intuitively, we expect a better design procedure to select slices that will yield better predictive ability, therefore allowing more efficient imputation of the "atlas". We compared our *EIG* experimental design method to the *Random* approach and to the *Serial* approach.

We found that our design procedure yielded improved predictive performance compared to the competing approaches (Fig. 2d).

Specifically, the $R^2$ of the predictions for the *EIG* design method approached the $R^2$ level of the complete atlas in fewer experimental iterations than competing approaches. The *EIG* method reached the performance of the full atlas after collecting roughly four slices, while the *Serial* method required roughly six slices to achieve comparable performance. This result suggests that the *EIG* approach selects cross-sections that allow for efficient construction of cell atlases.

**Border finding.** We next studied a simulated setting in which the goal is to identify the location and boundaries of a tissue region of interest. This simulation mimics several applications in spatial genomics, such as identifying the boundaries of a tumor and localizing an anatomical region of interest. We generated a dataset with two-dimensional spatial coordinates, similar to the data generated in Section 2.2. On the interior of the synthetic tissue, we designated points within a circle as the region of interest (ROI; Fig. 3a). The points outside of this circle were given a label of region of non-interest. We injected noise to these labels so that 10% of all points were mislabeled. The goal of this experiment was to collect cross-sections of points in order to localize the ROI as quickly as possible.

We applied our method to this dataset using the spherical border model (Eq. (13)). The parameters of interest in this model are the center and radius of the sphere; our approach thus maximizes the EIG in the posterior over these parameters.

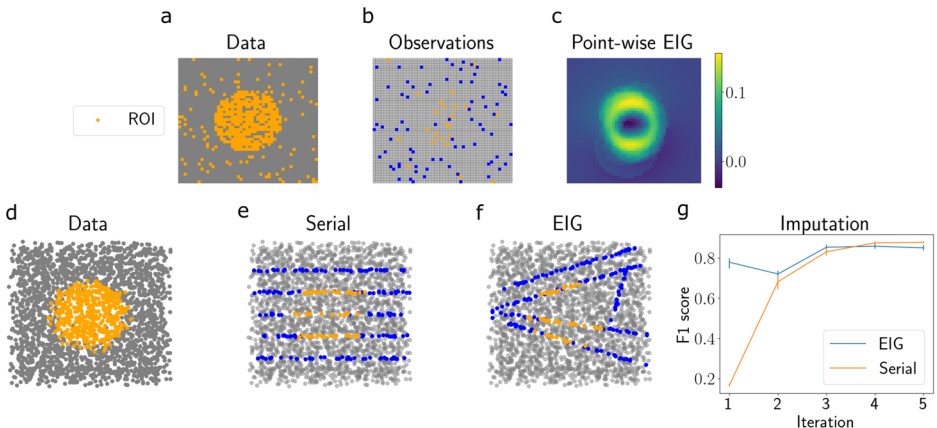

**Fig. 3 | Synthetic slicing experiment for localizing a region of interest. a** Two-dimensional simulated spatial gene expression data with a region of interest in orange (ROI). **b** Point-wise observations. Orange points are labeled as belonging to the ROI, blue points are outside the ROI, and gray points are unobserved. **c** Estimated expected information gain (EIG) for each spatial location, where each design is a single point. **d** Estimated EIG for each horizontal slice design. **e** Synthetic ROI data. **f** Slices chosen after $T = 5$ iterations of running our model. **g** Mean F1 score of predictions after each iteration. Error bars represent 95% confidence intervals computed using $n = 5$ runs. Source data are provided as a Source Data file.

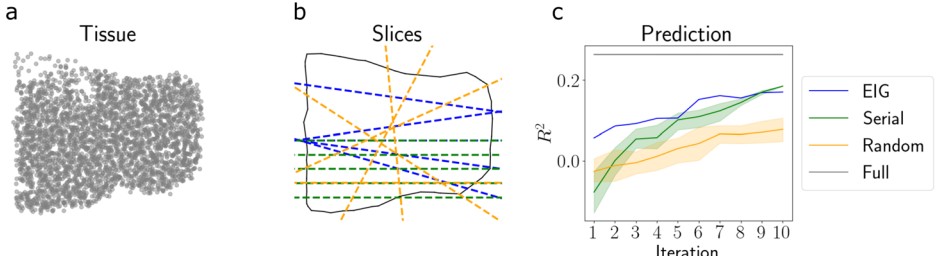

**Fig. 4 | Application to Visium data. a** Spatial locations of tissue. **b** Slices chosen by each approach after $T = 5$ iterations. The outline of the tissue is shown by the solid black line, and the slices chosen by each approach are shown by the dashed lines. The color legend is in panel (**c**). The full lines are drawn for clarity, but only the noninitersecting piece (within one tissue fragment) is considered as the relevant slice. **c** Predictive $R^2$ of the held-out gene expression for both approaches across iterations. Error bands represent 95% confidence intervals computed using $n = 5$ runs. Source data are provided as a Source Data file.

To demonstrate how the EIG objective behaves in this setting, we first used an artificial design space where each design was a single point rather than a cross-section. We randomly selected 100 points as observations and ran the *EIG* model for one iteration. We then visualized the EIG for each candidate design (each of which was a point in this case). We observed that the EIG was highest for points at the border of the ROI (Fig. 3c). This observation implies that, in order to learn the center and radius of the sphere, it is most informative to sample points near the estimated border.

We then extended this experiment to a design space where each design was a line. We ran the *EIG* and *Serial* approaches for $T = 5$ iterations. For the *Serial* strategy, we randomized the order in which the slices were chosen. After each iteration, we visualized the chosen slices, computed predicted labels (ROI or not-ROI) for each point, and computed the predictive performance using the F1 score. We found that the *EIG* approach obtained its maximum predictive performance in fewer iterations compared to the *Serial* approach (Fig. 3g). This result highlights the usefulness of our design approach when the goal is to localize a region of interest.

**Application to Visium data**

Next, we applied our experimental design approach to a series of spatial gene expression datasets. We first leveraged spatial transcriptomics data from the 10x Genomics Visium platform[21]. This dataset consists of a two-dimensional section from the sagittal-posterior region of a mouse brain. Since a full three-dimensional profile of the brain was not available for this dataset, we considered one-dimensional slices through this two-dimensional tissue as a proof

of concept. The goal of this experiment was to characterize the gene expression patterns across the tissue as thoroughly as possible; in other words, the goal was to build an atlas. Thus, we modeled the data with the atlas-building GP regression model (Eq. (5)), where the parameter of interest is the entire function $f$ governing the spatial organization of gene expression[22].

We ran the *EIG*, *Serial*, and *Random* design approaches for $T = 10$ iterations and visualized the resulting slices. For the *Serial* strategy, we randomize the ordering of the ten slices in each repetition of the experiment. We also sought to quantify the downstream utility of the chosen slices. To do so, we evaluated our ability to impute the gene expression levels at unobserved locations after collecting each slice. We used a GP with an RBF covariance function to make predictions.

We found that the EIG-maximizing slices tended to be the ones that covered the most surface area of the tissue and intersected the most spots, and these slices thoroughly covered the domain of the tissue (Fig. 4b). Moreover, we found that our approach achieved a higher imputation performance across experimental iterations compared to the competing approaches (Fig. 4c). This result suggests that our design procedure could be useful for selecting tissue cross-sections that will ultimately be used to construct an atlas of the entire tissue.

**Reconstructing the Allen Brain Atlas**

We next applied our experimental design method to three-dimensional spatial gene expression data from the mouse brain in the Allen Brain Atlas[2]. This dataset contains the expression levels of approximately 20,000 genes in the adult mouse brain and were

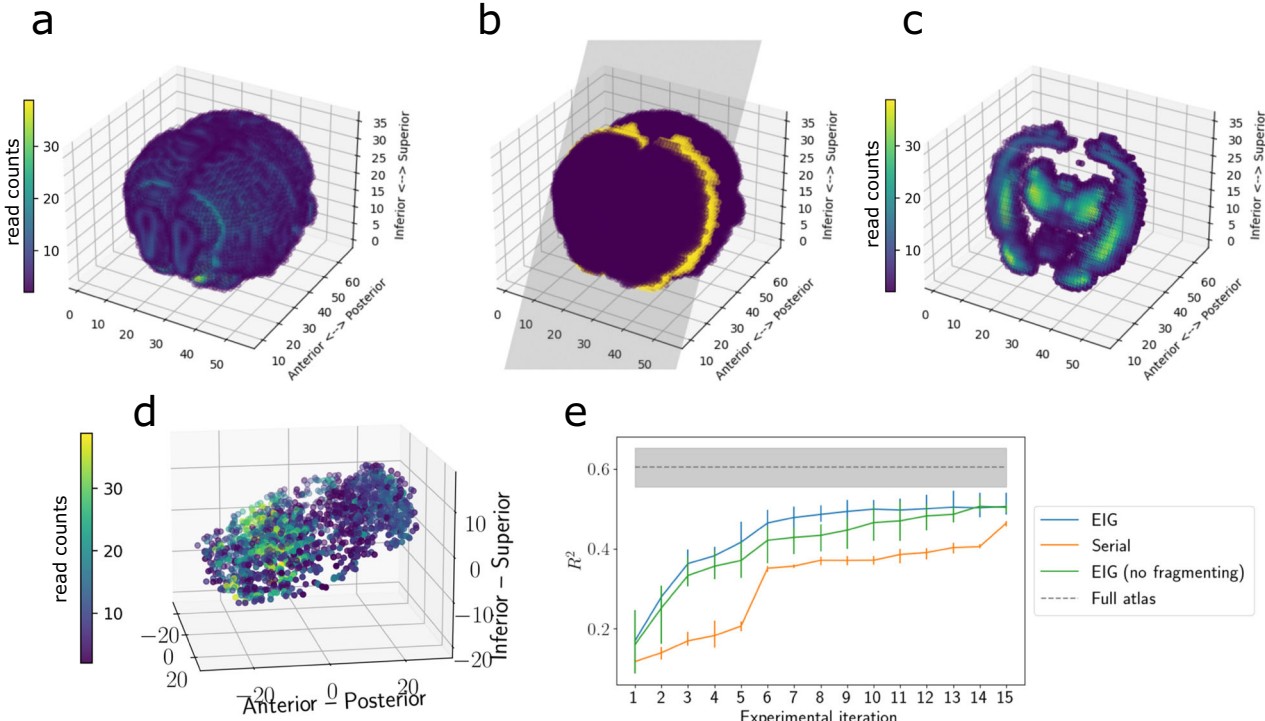

**Fig. 5 | Reconstructing the Allen Brain Atlas. a** Allen Brain Atlas coordinates colored by the expression of *PCP4*. **b** An example slice through the coordinates. **c** The resulting observations after taking this slice. **d** The slices and observations chosen by the *EIG* approach. **e** Imputation performance across experimental iterations. Error bars represent 95% confidence intervals computed using *n* = 5 runs. Source data are provided as a Source Data file.

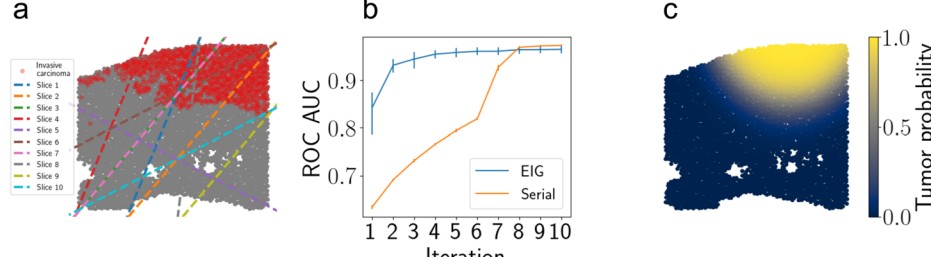

**Fig. 6 | Localizing invasive carcinoma in prostate tissue. a** Slices chosen by the *EIG* method. Cancerous spots are shown in red. The full lines are drawn for clarity, but only the nonintersecting piece (within one tissue fragment) is considered as the relevant slice. **b** F1 score of tumor/healthy label predictions after each iteration of experimental design. Error bars represent 95% confidence intervals computed using *n* = 5 runs. **c** Tumor/healthy predictions following five iterations of design. Stronger yellow color indicates spots with higher predicted probability of containing tumorous tissue. Source data are provided as a Source Data file.

collected using in situ hybridization (ISH). The data are collected as images where the pixel intensity encodes the level of gene expression at each spatial location. The data were collected sagittal sections of the mouse brain that were 200 $\mu$m apart from one another.

While these data were collected with serial slices of the tissue, we sought to answer whether an atlas of equal precision could be constructed with fewer slices using our design approach. To do this, we applied our slicing algorithm to the data and evaluated our ability to impute the gene expression levels of the full atlas. After each slice, we predict the gene expression levels at all unobserved locations and compute the prediction error. For comparison, we compared against two competing slicing strategies: one that takes serial sagittal slices and another that takes random slices. See Supplementary A.2 for details.

We found that we could reconstruct the atlas within reasonable accuracy with fewer samples than were taken in the original atlas (Fig. 5).

## Localizing invasive carcinoma in prostate tissue

As a final application of our experimental design approach, we considered the problem of localizing a tumor within a tissue by taking sequential slices of the tissue. We leveraged another spatial transcriptomics dataset from the 10x Visium platform that profiled a human prostate cancer sample[21]. The dataset contains a cross-section of the tissue with spatially-resolved transcriptomic data at each location, as well as pathologist annotations of the cancerous tissue region (Fig. 6a). We modeled the data with the spherical border model (Eq. (13)), where again we are interested in estimating the center and radius of the tumor.

We ran the design approaches forward for *T* = 10 iterations. On each iteration, we computed model predictions for whether each spatial location corresponded to tumorous or healthy tissue, and we used these predictions to compute the F1 classification score (Fig. 6c). We found that the classification performance increased rapidly as more slices were collected. This experiment demonstrates the

versatility of our design approach to extend to a setting in which a targeted region of the tissue is being localized.

## Discussion

In this paper, we formalized the problem of choosing slice locations for spatial genomics experiments, and we proposed a set of methods for performing experimental design in this setting. We focused on optimizing two study types in particular: constructing an atlas of an entire tissue, and localizing a particular region of a tissue. We applied our method to a range of synthetic datasets, a spatial gene expression dataset from the mouse cortex, and an ISH dataset from the Allen Brain Atlas. In each of these cases, we demonstrated the value of optimizing the locations of cross-sections. As spatial genomic profiling technologies evolve, we envision our approach being useful for planning data collection to be as efficient as possible.

Our work has several limitations that need to be solved before directly applying our method to design spatial genomics experiments. First, the tools used to obtain slices of tissues may not be precise enough to obtain an exact slice location that is prescribed by our approach. Second, our approach assumes that a model of the tissue's shape is available. While such a model is available for many tissue types, it may not be available for less well-studied tissue types. However, even a rough model of the tissue shape suffices for most applications of our model.

This work motivates several future directions. First, the experimental design problem could be extended to account for the different types of spatially-resolved measurement technologies available to an experimentalist on any given iteration, as well as their associated costs, spatial resolutions, and levels of precision. Second, there is an opportunity to further formalize the problem of designing experiments with highly structured design constraints, as well as propose new algorithms for optimizing the associated objective. In the current study, we took a simple, brute-force approach to searching over the candidate cross-sections, but more efficient search and optimization strategies could be explored. Third, while our method organically allows for batch designs, there may be methodological improvements to be made in a setting where multiple slices are taken on each iteration. Fourth, our experimental design method could be integrated with downstream analysis methods (e.g., cell-type deconvolution) when the downstream task is integral to the design of the study. Finally, there is an opportunity to tailor the models proposed in this work to count-based likelihoods. While we focus on Gaussian likelihoods in this work for simplicity and computational tractability, it is possible that using a Poisson or negative binomial distribution could yield improvements in downstream results. In particular, the proposed variational experimental design algorithm can be applied to, for instance, a Poisson Gaussian process model.

## Methods

### Notation and problem statement

We now formalize the problem studied in this paper. We consider a spatial genomics dataset consisting of pairs $(\mathbf{x}, \mathbf{y})$, where $\mathbf{x} \in \mathcal{X}$ is a spatial location, $\mathcal{X} \subset \mathbb{R}^3$ is the spatial domain of the tissue (we are assuming it is three dimensional), and $\mathbf{y}$ is a $p$-dimensional outcome at this location (e.g., a vector of gene expression, protein expression, or another univariate or multivariate phenotype at this location). We denote a dataset of $n$ such pairs as $\{(\mathbf{x}_i, \mathbf{y}_i)\}_{i=1}^n$. In matrix form, we define $\mathbf{X}_t$ and $\mathbf{Y}_t$ to be the $n \times 3$ and $n \times p$ matrices whose $i$th rows are the $\mathbf{x}_i$ and $\mathbf{y}_i$, respectively.

Suppose the goal of an experiment is to profile the phenotype of a tissue, organ, or entire organism whose cells lie in the spatial domain $\mathcal{X}$. Spatially-resolved genomic data is typically collected from *slices* of the tissue. Each slice is a two-dimensional cross-section of the 3D coordinate system $\mathcal{X}$. Let $P$ represent a plane intersecting the tissue. The set of points on a cross-section defined by $P$ is given by $\mathcal{X} \cap P$.

In practice, tissue sections have nonzero width, meaning that points nearby a cross-section's plane will also be observed. For a slice with half-width $\delta$, we denote the set of points observed by a cross-section defined by plane $P$ as $\mathcal{X}_{P_\delta}$; these are the spatial locations within a Euclidean distance of $\delta$ to the plane $P$.

We consider the class of iterative experimental strategies, where spatial genomics readouts are collected in $T$ sequential batches, and each batch is a single experiment that collects data from one tissue slice. When planning batch $t \in [T]$, our goal is to select a cross-section defined by $P_t$ that maximizes the expected utility gained from performing that experiment. Let $\mathcal{P}$ denote the set of candidate planes, and let $\theta \in \Theta$ be a set of unknown model parameters. Throughout this paper, $\mathcal{P}$ will constitute our *design space*, or the set of designs to be chosen from. We define $\mathcal{U} : \mathcal{P} \times \Theta \times \mathcal{Y} \to \mathbb{R}$ to be a utility function that, intuitively, measures the goodness of an experiment and its expected observations. We discuss the choice of the utility function in the next section. On the first batch, the optimal slice is given by

$$P_1^\star = \arg\max_P \; \mathbb{E}_{\theta, \mathbf{Y}_1} \left[ \mathcal{U}(P, \theta, \mathbf{Y}_1) \right], \tag{3}$$

where $\mathbf{Y}_1$ is the set of outcomes observed on the first batch. The maximizer for batches $t > 1$ is similar but also relies on the data collected up to that point to inform the design of the current batch. For $t > 1$ the solution is

$$P_t^\star = \arg\max_P \; \mathbb{E}_{\theta, \mathbf{Y}_t} \left[ \mathcal{U}(P, \theta, \mathbf{Y}_{1:t-1}, \mathbf{Y}_t) \right], \tag{4}$$

where $\mathbf{Y}_t$ are the outcomes observed on iteration $t$ and $\mathbf{Y}_{1:t} = \{\mathbf{Y}_1, \ldots, \mathbf{Y}_t\}$.

While this approach assumes a single slice is assayed in each iteration, our approach trivially extends to *minibatch* mode, where $m$ slices are collected and assayed in a single iteration and before the model is updated; this scenario is the most likely experimentally. An important aspect of this design problem—and one that makes it unique from related experimental design problems—is its highly structured design space. Specifically, while other design problems allow the experimentalist to freely choose one or multiple designs (unique values of $\mathbf{x}$) on each iteration, the spatial slicing problem requires that the spatial locations be situated on the same plane. We refer to this general problem class as *structured batch experimental design*, which encompasses experimental settings where one or more destructive and constrained tissue slices are collected on each iteration. In our case, the samples are constrained to lie on a plane. Other fields in which structured batch experimental design might appear are tomography[23] and pathology[24].

We now describe our approach to the spatial experimental design problem. We consider two possible experimental goals:

1. Building a spatially-resolved genomic atlas for a tissue or organ;
2. Localizing a tissue region of interest, such a tumor or anatomical region.

For each of these applications, we formalize the experimental goal, and we propose a statistical model and utility function that reflect the associated goal. We then propose an optimization scheme to iteratively find the sample cross-section with maximum expected utility.

### Atlas building objective

A long-term goal in genomics and biology is to build a comprehensive characterization of all cell types in the human body. This is commonly referred to as an *atlas*, which draws an analogy with a "map" of cells' physical organization and phenotypes[25]. In its ideal form, a comprehensive atlas would allow researchers to query the atlas using a spatial location or region of interest, and the query would return a detailed description of the phenotype, including cell types and cell states.

Atlases for a set of human and mouse tissue types have been constructed using various data modalities, such as in situ hybridization, single-cell RNA-sequencing, histology, and spatial gene expression[2,25–28]. However, comprehensive atlases using the most modern spatial gene expression profiling methods have yet to be established.

Here, we consider the problem of efficiently constructing an atlas using spatial genomics technologies. We first formalize the problem of building a spatially-resolved atlas and then discuss our proposed approach. For simplicity, we first consider a noisy univariate phenotype $y$ (e.g., the expression level of one gene) and move to multivariate phenotypes later. Suppose $y$ follows a spatial process defined on the domain $\mathcal{X} \subset \mathbb{R}^3$. Consider the following model[22]:

$$y = f(\mathbf{x}) + \epsilon, f \sim \mathrm{GP}(0, k(\cdot, \cdot)), \epsilon \sim N(0, \tau^2), \tag{5}$$

where $f$ has a Gaussian process prior with mean zero and covariance function $k(\cdot, \cdot)$, and $\epsilon$ is Gaussian noise with variance $\tau^2$. Under this model, the unknown function $f(\cdot)$ provides a full description of the spatially-resolved atlas for phenotype $y$. In particular, given spatial location $\mathbf{x}$, the function evaluation $f(\mathbf{x})$ tells us the (noiseless) value of $y$ at that location. Thus, the statistical goal in atlas-building is to infer $f(\mathbf{x})$ for every location $\mathbf{x}$ within the 3D domain. We take a Bayesian approach to this problem, where estimating the function $f(\cdot)$ amounts to computing the posterior distribution for $f$ given the collected data,

$$p(f | \{(\mathbf{x}_i, \mathbf{y}_i)\}_{i=1}^n) = \frac{p(\{(\mathbf{x}_i, \mathbf{y}_i)\}_{i=1}^n | f) p(f)}{p(\{(\mathbf{x}_i, \mathbf{y}_i)\}_{i=1}^n)}. \tag{6}$$

Our experimental design objective is then to choose tissue slices that are expected to maximally "improve" this posterior distribution in some way. We discuss metrics to quantify this improvement next.

**Atlas construction via information gain.** Since our ultimate goal in the atlas-building objective is to infer the unobserved function $f$, we choose a utility function that rewards experimental designs that offer more information about $f$. The *information gain* (IG) is a utility function defined as the difference in entropy between the prior and the posterior distributions. Under our atlas model, the IG of taking a slice defined by plane $P_t$ and observing outcome $\mathbf{Y}_t$ is $\mathrm{IG}(\mathbf{Y}_t, P_t) = H[p(f | \mathbf{Y}_{1:t-1})] - H[p(f | \mathbf{Y}_{1:t}, P_t)]$, where $H[\cdot]$ is the differential entropy functional. Because $\mathbf{Y}_t$ is not observed before conducting experiment $t$, we cannot directly optimize the IG with respect to $P_t$. Thus, we take the expectation of the IG with respect to $\mathbf{Y}_t$, which is a quantity known as the *expected information gain* (EIG). The EIG of design $P_t$ is given by

$$\mathrm{EIG}(P_t) = \mathbb{E}_{\mathbf{Y}_t, f} \left[ \log p(\mathbf{Y}_t | \mathcal{D}_{t-1}, P_t) - \log p(\mathbf{Y}_t | f, \mathcal{D}_{t-1}, P_t) \right], \tag{7}$$

where $\mathcal{D}_{t-1} := \{(\mathbf{X}_{1:t-1}, \mathbf{Y}_{1:t-1})\}$ represents the data observed through experimental iterations $1, \ldots, 1 - t$, and $\mathbf{X}_{1:t}$ and $\mathbf{Y}_{1:t}$ are the spatial locations and associated outcomes, respectively. We then maximize the EIG (Eq. (7)) with respect to $P_t$.

Under our GP regression atlas model (Eq. (5)), the EIG can be computed analytically. Let $\mathbf{X}_t$ be the set of spatial locations captured on the cross-section defined by plane $P_t$. The EIG for $P_t$ is

$$\mathrm{EIG}(P_t) = \frac{1}{2} \log \det \left( \frac{1}{\tau^2} \widehat{\Sigma}(\mathbf{X}_t) + \mathbf{I} \right), \tag{8}$$

where the predictive covariance $\widehat{\Sigma}(\mathbf{X}_t)$ of the GP at locations $\mathbf{X}_t$ is given by

$$\widehat{\Sigma}(\mathbf{X}_t) = \mathbf{K}_{\mathbf{X}_t \mathbf{X}_t} - \mathbf{K}_{\mathbf{X}_t \mathbf{X}_{t-1}} (\mathbf{K}_{\mathbf{X}_{1:t-1} \mathbf{x}_{1:t-1}} + \tau^2 \mathbf{I})^{-1} \mathbf{K}_{\mathbf{X}_{1:t-1} \mathbf{x}_t}. \tag{9}$$

We use the notation $\mathbf{K}_{\mathbf{X}\mathbf{X}'}$ to denote the matrix of covariance function evaluations whose $ij$th element is $[\mathbf{K}_{\mathbf{X}\mathbf{X}'}]_{ij} = k(\mathbf{x}_i, \mathbf{x}'_j)$. See Supplementary A.1.2 for a full derivation of these quantities. Our optimization problem under the atlas-building objective is to maximize the EIG with respect to $P_t$.

**Maximizing information gain to find the optimal cross-section.** Our goal is to find the set of points in a tissue that maximize the EIG (Eq. (8)). However, because we are constrained to collect two-dimensional cross-sections of a tissue, rather than any arbitrary subset of spatial locations, we must constrain each candidate design's spatial locations $\mathbf{X}$ to lie on a plane.

This is equivalent to choosing a plane $P_t$ representing a two-dimensional cross-section of the slice collected at time $t$. Although there are an infinite number of potential cross-sections, we simplify the optimization problem by discretizing the space of cross-sections. Specifically, we create a design space with $D$ cross-sections, where $D$ can be chosen depending on computational resources and required precision. The optimization problem then reduces to maximizing over a discrete set, which in this setting is typically a tractable problem.

Because slices consist of entire cross-sections of the tissue, each slice creates a new, disjoint tissue fragment. At the start of iteration $t$, we will have taken $t - 1$ slices, so the tissue will be split into $t$ disjoint fragments. We account for this by considering the candidate cross-sections in each fragment separately. Let $\mathcal{X}^t$ be the set of spatial locations corresponding to tissue fragment $t$, and let $\mathcal{P}_t$ be the set of planes intersecting $\mathcal{X}_t$. Note that, by definition, for all $t$ it must hold that

$$\mathcal{X}^1 \equiv \mathcal{X}, \quad \bigcup_{i=1}^t \mathcal{X}^i = \mathcal{X}, \quad \text{and} \bigcap_{i=1}^t \mathcal{X}^i = \emptyset. \tag{10}$$

The optimization problem on iteration $t$ is then to find the slice (or $m$ slices) that maximizes the EIG for the current set of tissue fragments: $P_t^\star = \arg\max_{P \in \{\mathcal{P}_i\}_{i=1}^t} \mathrm{EIG}(P)$.

## Localizing a tissue region of interest

Next, we consider an experimental design setting in which there is a particular region of space within $\mathcal{X}$ whose borders we would like to identify. For example, we may be studying a biopsy sample from a tumor tissue, and we would like to identify the border between the tumor and healthy tissue using as few slices as possible. This localization problem is a common goal in cancer pathology[29].

For simplicity, assume that we can label each spatial location as being on the interior of the region of interest ($y = 1$) or the exterior of the region of interest ($y = 0$) after we have collected data for that location. On experimental iteration $t$, we collect data from a cross-section of the tissue defined by $P_t$, where the data are made up of the spatial locations $\mathbf{X}_t \in \mathbb{R}^{n_t \times 3}$ and the interior/exterior labels for those locations $\mathbf{y}_t \in \{0, 1\}^{n_t}$, where $n_t$ is the number of points observed after taking slice $P_t$.

**Bounding box model.** Consider a logistic regression model where the area of interest is modeled with an axis-aligned rectangular bounding box. Assume a spatially varying Bernoulli likelihood, $y \sim \mathrm{Bern}(g(\mathbf{x}))$, where $g(\cdot)$ is a link function mapping the spatial coordinates to the bounding box probability model. For the axis-aligned bounding box, we parameterize $g(\cdot)$ as follows:

$$g(\mathbf{x}) = \prod_{d=1}^3 \left[ \left(1 + \exp\{-(\mathbf{x}^\top \mathbf{e}_d + c_d + a_d)\}\right) \left(1 + \exp\{\mathbf{x}^\top \mathbf{e}_d + c_d - a_d\}\right) \right]^{-1}, \tag{11}$$

where $\mathbf{c}, \mathbf{a} \in \mathbb{R}^3$ are parameters controlling the center and width of the box, respectively, and $\mathbf{e}_d$ is the $d$th axis-aligned unit vector of

length 3. Isotropic Gaussian priors can be used, i.e., $\mathbf{a}, \mathbf{c} \sim N(\mathbf{0}, \mathbf{I})$. This model, which is a generalization of a logistic regression model, captures the borders of the region of interest through the parameters $\theta = \{\mathbf{c}, \mathbf{a}\}$. Thus, the posterior after iteration $t$ is $p(\theta | \mathbf{X}_{1:t}, \mathbf{y}_{1:t})$. Recall that $\mathcal{D}_{t-1} := \{(\mathbf{X}_{1:t-1}, \mathbf{y}_{1:t-1})\}$ represents the data observed through experimental iterations $1, \ldots, t-1$. The expected information gain for a slice through plane $P_t$ is

$$\mathrm{EIG}(P_t) = \mathbb{E}_{\theta, \mathbf{y}_t | \mathcal{D}_{t-1}, \mathbf{X}_t} \left[ H[p(\theta|\mathcal{D}_{t-1})] - H[p(\theta|\mathcal{D}_{t-1}, P_t, \mathbf{y}_t)] \right]. \quad (12)$$

In order to select the slices to identify the borders of the region of interest representing the tumor, we maximize the EIG with respect to $P_t$.

**Spherical and elliptical border model.** We may parameterize the border of a region of interest using shapes other than a rectangle. For example, we may use a circular bounding area instead. Recall that the equation for the points in $\mathcal{X}$ contained within a ball with center $\mathbf{c}$ and radius $r$ is given by $\{\mathbf{x} \in \mathcal{X} : \| \mathbf{x} - \mathbf{c} \|_2 \le r\}$. A viable statistical model is then

$$y \sim \mathrm{Bern}(g(\mathbf{x})), \quad g(\mathbf{x}) = (1 + \exp\{\|\tilde{\mathbf{x}} - \mathbf{c}\|_2 - r\})^{-1}, \quad \mathbf{c} \sim \pi_c, \quad r \sim \pi_r, \quad (13)$$

where $\pi_c$ and $\pi_r$ are prior distributions for the center and radius, respectively.

The spherical border model can also be generalized to an elliptical border. Recall that an ellipsoid can be written as a linear transformation of a sphere. The points contained inside the ellipsoid are $\{\mathbf{x} \in \mathcal{X} : \|\mathbf{A}\mathbf{x} - \mathbf{c}\|_2^2 = r^2\}$, where $\mathbf{A} \in \mathbb{R}^{3 \times 3}$. These options for border-finding extend our experimental design approach.

We briefly review the NMC estimator to solve this problem. Expanding the EIG expression, we see that it contains a nested integral:

$$
\begin{aligned}
\mathrm{EIG}(P_t) &= \mathbb{E}_{\mathbf{y}_t, \theta} \left[ \log \frac{p(\mathbf{y}_t | \theta, \mathcal{D}_{t-1}, P_t)}{p(\mathbf{y}_t | \mathcal{D}_{t-1}, P_t)} \right] \\
&= \int_{\mathcal{Y} \times \Theta} p(\mathbf{y}_t, \theta | P_t, \mathcal{D}_{t-1}) \log \frac{p(\mathbf{y}_t | \theta, \mathcal{D}_{t-1}, P_t)}{p(\mathbf{y}_t | \mathcal{D}_{t-1}, P_t)} \, d\mathbf{y}_t \, d\theta \\
&= \int_{\mathcal{Y} \times \Theta} p(\mathbf{y}_t, \theta | P_t, \mathcal{D}_{t-1}) \log \frac{p(\mathbf{y}_t | \theta, \mathcal{D}_{t-1}, P_t)}{\int_{\Theta} p(\theta' | \mathcal{D}_{t-1}) p(\mathbf{y}_t, | \theta', \mathcal{D}_{t-1}, P_t) d\theta'} \, d\mathbf{y}_t \, d\theta.
\end{aligned}
\quad (14)
$$

The NMC estimator[30] approximates these nested integrals using a Monte Carlo estimator.

Taking $S$ samples for the outer sum and $S'$ samples for the inner sum, the NMC approximation is

$$\widehat{\mathrm{EIG}}(P_t) := \frac{1}{S} \sum_{s=1}^{S} \log \left\{ \frac{p(\mathbf{y}_t^{(s)} | \theta^{(s,0)}, \mathcal{D}_{t-1}, P_t)}{\frac{1}{S'} \sum_{s'=1}^{S'} p(\mathbf{y}_t^{(s)} | \theta^{(s,s')}, \mathcal{D}_{t-1}, P_t)} \right\}, \quad (15)$$

where $\theta^{(s,0)}$ and $\theta^{(s,s')}$ are sampled from a variational approximation to the posterior for $\theta$, and $\mathbf{y}_t^{(s)}$ is sampled from the likelihood model:

$$\theta^{(s,0)}, \theta^{(s,s')} \sim q_{t-1}(\theta), \quad \mathbf{y}_t^{(s)} \sim p(\mathbf{y}_t | \theta = \theta^{(s,0)}, P_t). \quad (16)$$

Here, $q_{t-1}(\theta)$ is a variational approximation to the posterior $p(\theta | \mathcal{D}_{t-1})$ (see Supplementary A.1.3 for details). Algorithm 1 contains a detailed exposition of the approach. Increasing the number of Monte Carlo samples $S, S'$ trades off computation speed for a more precise estimate of the EIG. Selecting the optimal slice is then performed via $P_t^* = \arg\max_P \widehat{\mathrm{EIG}}(P)$.

---

**Algorithm 1.** Nested Monte Carlo sampling on experimental iteration $t$ for design $P_t$

1 **Input**: Data $\mathcal{D}_{t-1} = \{(\mathbf{X}_{1:t-1}, \mathbf{Y}_{1:t-1})\}$ and candidate design $P_t$.
2 Compute variational approximation $q_{t-1}(\theta) \approx p(\theta | \mathcal{D}_{t-1})$ (Supplementary A.1.3);
3 **for** $s = 1, \ldots, S$ **do**
4 Sample $\theta^{(s,0)} \sim q_{t-1}(\theta)$;
5 Sample $\mathbf{y}_t^{(s)} \sim p(y | \theta = \theta^{(s,0)}, P_t)$;
6 **for** $s' = 1, \ldots, S'$ **do**
7 Sample $\theta^{(s,s')} \sim q_{t-1}(\theta)$;
8 **end**
9 **end**
10 Plug samples $\left\{ \left( \theta^{(s,0)}, \mathbf{y}_t^{(s)}, \{\theta^{(s,s')}\}_{s'=1}^{S'} \right) \right\}_{s=1}^{S}$ into Equation (15) to estimate EIG.

---

**Inference.** Under any of these border-finding models, our goal is to find the design that maximizes the EIG ((7)). For most modeling settings, there are three intractable quantities in the expression for EIG: the posterior $p(\theta | \mathcal{D}_{1:t-1})$, the entropy of the posterior, and the outer expectation over the data.

We use a two-step approximation for these intractable quantities. First, while designing iteration $t$, we compute an approximation to the posterior $p(\theta | \mathcal{D}_{1:t-1})$ using variational inference. We denote this approximation as $q_{t-1}(\theta)$ (see Supplementary A.1.3 for details). Second, we use this approximate posterior to estimate the EIG using a nested Monte Carlo (NMC) sampling approach[30].

**Statistics and reproducibility**
In this study, we did not use traditional statistical tests as part of our analysis. Despite the absence of statistical testing, our work rigorously presents data with appropriate measures of variability. Each plot in our study includes error bars or confidence intervals, clearly labeled to represent the variability or uncertainty in the data.

**Reporting summary**
Further information on research design is available in the Nature Portfolio Reporting Summary linked to this article.

## Data availability

All relevant data supporting the key findings of this study are available within the article and its Supplementary Information files. Below we include a description of each data source. *Synthetic data.* Code for generating synthetic datasets can be found in the GitHub repository: https://github.com/andrewcharlesjones/spatial-experimental-design. *Visium mouse brain data.* Visium data were obtained from the 10x Genomics website. Data for the one slice was downloaded from the "Datasets" page. Specifically, spatial gene expression was downloaded from the page called https://www.10xgenomics.com/resources/datasets/mouse-brain-serial-section-1-sagittal-posterior-1-standard-1-1-0 *Mouse Brain Serial Section 1 (Sagittal-Posterior). Allen Brain Atlas data* The in situ hybridization data from the Allen Brain Atlas was downloaded using the `brainrender` Python API[31]. The GitHub repository for the API is available at https://github.com/brainglobe/brainrender. We used data from the study with experiment ID 79912613. *Visium prostate cancer data.* Visium data were obtained from the 10x Genomics website from the Datasets page titled https://www.10xgenomics.com/resources/datasets/human-prostate-cancer-adenocarcinoma-with-invasive-carcinoma-ffpe-1-standard-1-3-0 *Human Prostate Cancer, Adenocarcinoma with Invasive Carcinoma (FFPE).* Source data are provided with this paper.

## Code availability

Code for all data preprocessing, synthetic data generation, and experiments can be found in the GitHub repository: github.com/andrewcharlesjones/spatial-experimental-design[32].

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

## Acknowledgements

The authors thank Addie Minerva, Cate Peña, Bianca Dumitrascu, and Geoffrey Roeder for helpful conversations. AJ and BEE were funded by Helmsley Trust grant AWD1006624, NIH NCI 5U2CCA233195, NIH NHGRI R01 HG012967, a CZI Data Insights grant, and NSF CAREER AWD1005627. BEE is a CIFAR Fellow in the Multiscale Human Program. DC was supported in part by a Google Ph.D. Fellowship in Machine Learning. DL was supported by NIH NCATS UL1 TR002489, NIH NHLBI R01 HL149683, NIH NIEHS P30 ES010126, and NIH NLM R56 LM013784.

## Author contributions

A.J., D.C., D.L., and B.E.E. designed the method. A.J. implemented the method and conducted data analysis. A.J., D.C., D.L., and B.E.E. analyzed the results. A.J., D.C., D.L., and B.E.E. wrote the manuscript.

## Competing interests

B.E.E. is on the SAB of Creyon Bio, Arrepath, and Freenome. B.E.E. is a consultant with Neumora. The remaining authors declare no competing interests.
