## [Peer Review File · Nature Communications]

Optimizing the design of spatial genomic studiesReviewer #1 (Remarks to the Author):

In this paper, the authors proposed structured batch experimental design, a novel approach to profile tissue cross sections for spatial genomic studies in a cost effective way. They considered two experimental goals, including (i) building a spatially-resolved genomic atlas for a tissue or organ; and (ii) localizing a tissue region of interest. Overall this study is timely given the increasing interest and need for designing 3D spatial profiling, atlas construction and tissue compartment identification using spatial genomics technologies. The proposed approach is based on well-established and appropriate theoretical rationale and rigorous in general. However, I have some concerns regarding how the proposed algorithm is evaluated and compared with competing approaches. Here I provided several comments for clarification.

1. Figure 2d. Although it was claimed that EIG outperforms Serial, if we consider intervals of each method (color band), differences among methods are essentially negligible. Can authors quantify and justify gains of using EIG compared to competing approaches?

2. Figure 3. In Figure 3g, F1 scores for EIG and Serial are essentially identical except iteration 1. So it is not clear whether the difference between EIG and Serial is really meaningful here. I guess that in the case of Serial, the authors might first generate the top or the bottom line of Figure 3e in the first iteration. If this is really the case, that line misses the ROI totally and I believe that's why Serial has such a low F1 score in the first iteration and follows up EIG from the second iteration. What if the line in the middle of Figure 3e is chosen in the first iteration for Serial? I think that in this case, F1 scores for EIG and Serial will be identical across all the iterations. In reality, experimenters do not simply pick up a slice but rather decide one based on either H&E images or prior biological knowledge. Hence, I believe such way to pick up slides might be close to the reality. Can the authors discuss this issue in detail and consider other orders to pick up slices?

3. Figure 4. In Figure 4b, Serial lines are located only in the bottom side of the tissue, rather than distributed uniformly on the tissue. I believe Figure 4 shows a posterior region of mouse brain. In the case of posterior region of mouse brain, the right upper side of the tissue has the significantly complex segment structure compared to the left side or the bottom. Hence, covering the right upper side of the tissue will be critical. Again, as argued in Comment #2, experimenters usually pick up slices based on either H&E images or prior biological knowledge. Given this, it is more likely to put a line in the middle in the first iteration, one on the top quartile in the second iteration, one at the bottom in the third iteration, and putting the remaining ones in the middle of these lines. Can the authors provide the results when Serial is implemented in this way?

Reviewer #2 (Remarks to the Author):

The manuscript proposes an interesting approach for designing spatial genomic studies by sliding tissue samples. The proposed method has been demonstrated to have good theoretical properties and potential advantages in increasing information content using both simulation studies and real data examples. However, I have two major concerns:

Firstly, the proposed method of sliding tissue samples in spatial genomic studies is not practical for several reasons:

1. In current spatial genomic studies, a series of slides are cut from the tissue and processed through a pipeline of steps: 1. cutting the tissue into a series of slides, 2. sample preparation, 3. RNA extraction, 4. sequencing, and 5. data analysis. Step 1~3 are usually done within a short period of time (normally several hours) to avoid sample degradation. However, with the proposed method, cutting only one slide from the tissue and processing it through the entire pipeline (steps 1-5) would take at least a week. This is because the proposed method requires waiting for the results of the first cut before deciding how to make the next cut and repeating the pipeline. This would introduce unnecessary complications and delay the progress of the study.

2. Furthermore, the proposed method is likely to compromise the quality of the samples and the data generated from the study. After the first cut, if the sample is frozen and stored for a week, and need to be thawed before the next cut. The sample quality will degrade. This will introduce unnecessary damage to the sample and compromise the quality of the data generated from the study. Therefore, the proposed method is not only impractical but also likely to compromise the

quality of the samples and the data.

Secondly, the manuscript uses localizing a tissue region of interest as an example to demonstrate the advantage of the proposed method in designing spatial genomics studies. However, localizing tissue regions of interest is normally done by H&E stained tissue slides, which are much faster and cost only a few dollars per slide. Thus, it is not clear why spatial genomic experiments (which cost thousands of dollars) would be used for this purpose.

In summary, while the proposed method has good theoretical properties, it is not practically useful for real spatial genomic studies. Additionally, the manuscript's example of localizing tissue regions of interest using spatial genomics experiments is not convincing, given the availability of cheaper and faster alternatives.

Reviewer #3 (Remarks to the Author):

Designing a cost-efficient model by slicing a tissue that can maximize information gain with fewer slices is important. This study proposed an expected information gain (EIG) model based on the expected difference in entropy between the prior and posterior to optimize the tissue slicing. They validated their model by addressing targets including region of interest localization and gene expression characterization in a tissue. The manuscript reveals an interesting methodology for spatially-resolved genomic analysis and the results are well presented. The reviewer has the following concerns:

1. For spatial transcriptomics analysis, a more critical goal is to identify spatial domains and characterize the gene expression patterns within each domain. In particular, for datasets like 10X Visium, because each spot is a mixture of multiple cells, cell type deconvolution within spots or the identification of the distributions of each cell type within tissue is also significant for ST analysis. Therefore, rather than simply characterize the gene expression patterns across the entire tissue or identify the tissue borders, the authors can test whether their model could obtain complex domain information with fewer slices.
2. From the EIG algorithm, we can see that each slice is iteratively selected based on the information gained from the previous slices. As a comparison, the authors selected the "Serial" method (taking serial parallel cross sections along an anatomical plane) as a comparison. However, these parallel slices seem randomly selected in "Serial" method. It is unfair to make a comparison between the two methods. Similar to EIG model, the "Serial" method can also use the previously gained information to determine the next slice. For example, we can select the two adjacent slices which contain the largest information differences, and then use the slice in the middle of the two slices as the next slice. This strategy is widely used in many studies. I think the authors should compare their EIG model with the new "Serial" strategy I mentioned here instead of simply randomly selecting parallel slices.
3. In Figure 1e, Figure 3f, Figure 4b and Figure 6b, there are many cross slices within a tissue. This is not rational for a real experimental design for tissue segmentation.
4. Usually, the distributions of gene expression are followed by Poisson [PMID: 33603203] or negative binomial [PMID: 35449415, 35027729, 33037292, 34686758]. However, this study used a Gaussian distribution to describe gene expression. I don't think this assumption is rational for real spatial transcriptomics samples.
5. I don't know how to obtain the final fragments (i.e. Figure 1g) from the highest EIG slices (i.e. Figure 1e).
6. "We randomly selected 100 points as observations and ran the EIG model for one iteration." Can you explain why you need to do this step to run EIG?
7. Suggestion: In terms of Section 4.2, the cell distributions within the three simulated tissues are all uniformly distributed. I think the authors can test their EIG model in a more extreme situation, where cells are highly dense in some locations whereas they are very sparse in some other locations.

Response to comments on *Optimizing the design of spatial genomic studies*

Andrew Jones¹, Diana Cai¹, Didong Li², and Barbara E. Engelhardt^{3,4}

¹*Department of Computer Science, Princeton University*

²*Department of Biostatistics, University of North Carolina at Chapel Hill*

³*Gladstone Institutes*

⁴*Department of Biomedical Data Science, Stanford University*

Thank you for the thoughtful reviews. All of our responses and modifications to the main text are noted inline point-by-point below.

1 Reviewer 1

1.1 Remarks to the Author

In this paper, the authors proposed structured batch experimental design, a novel approach to profile tissue cross sections for spatial genomic studies in a cost effective way. They considered two experimental goals, including (i) building a spatially-resolved genomic atlas for a tissue or organ; and (ii) localizing a tissue region of interest. Overall this study is timely given the increasing interest and need for designing 3D spatial profiling, atlas construction and tissue compartment identification using spatial genomics technologies. The proposed approach is based on well-established and appropriate theoretical rationale and rigorous in general. However, I have some concerns regarding how the proposed algorithm is evaluated and compared with competing approaches. Here I provided several comments for clarification.

1.2 Major comments

1. Figure 2d. Although it was claimed that EIG outperforms Serial, if we consider intervals of each method (color band), differences among methods are essentially negligible. Can authors quantify and justify gains of using EIG compared to competing approaches?

Author response: Thank you for this comment. Our intention with Figure 2d was to demonstrate the performance of several variations of our proposed method

— EIG, EIG (no fragmenting), and EIG (parallel) — relative to the current baseline method (Serial). We find that a subset of these methods achieve higher imputation performance faster than the Serial approach, even after accounting for the confidence intervals. We have added a table (Supplementary Table 1) with the results to make the interpretation of the confidence intervals clearer.

2. Figure 3. In Figure 3g, F1 scores for EIG and Serial are essentially identical except iteration 1. So it is not clear whether the difference between EIG and Serial is really meaningful here. I guess that in the case of Serial, the authors might first generate the top or the bottom line of Figure 3e in the first iteration. If this is really the case, that line misses the ROI totally and I believe that's why Serial has such a low F1 score in the first iteration and follows up EIG from the second iteration. What if the line in the middle of Figure 3e is chosen in the first iteration for Serial? I think that in this case, F1 scores for EIG and Serial will be identical across all the iterations. In reality, experimenters do not simply pick up a slice but rather decide one based on either H&E images or prior biological knowledge. Hence, I believe such way to pick up slides might be close to the reality. Can the authors discuss this issue in detail and consider other orders to pick up slices?

Author response: Thank you for this comment. There may be some misunderstanding of the way we generate the *Serial* slices. Rather than always starting the slices from the top or bottom of the tissue, the order of the slices is chosen randomly. The slices themselves are always parallel, equidistant slices, but the ordering is not always sequential. The error bands for the *Serial* strategy in Fig. 3 incorporate randomly selected orderings of slice choices already. We have clarified this in the text in Section 4.2.3.

3. Figure 4. In Figure 4b, Serial lines are located only in the bottom side of the tissue, rather than distributed uniformly on the tissue. I believe Figure 4 shows a posterior region of mouse brain. In the case of posterior region of mouse brain, the right upper side of the tissue has the significantly complex segment structure compared to the left side or the bottom. Hence, covering the right upper side of the tissue will be critical. Again, as argued in Comment #2, experimenters usually pick up slices based on either H&E images or prior biological knowledge. Given this, it is more likely to put a line in the middle in the first iteration, one on the top quartile in the second iteration, one at the bottom in the third iteration, and putting the remaining ones in the middle of these lines. Can the authors provide the results when Serial is implemented in this way?

Author response: Thank you for this comment. In Fig. 4, we randomize the order of the Serial slices (this is what the confidence intervals are representing). Fig. 4b is simply a demonstration of each approach, but the ordering of the slices

is randomized across repetitions for the results in Fig. 4c. We have updated the text to clarify this.

2 Reviewer 2

2.1 Remarks to the author

The manuscript proposes an interesting approach for designing spatial genomic studies by sliding tissue samples. The proposed method has been demonstrated to have good theoretical properties and potential advantages in increasing information content using both simulation studies and real data examples. However, I have two major concerns.

2.2 Major comments

1. Firstly, the proposed method of sliding tissue samples in spatial genomic studies is not practical for several reasons:

- (a) In current spatial genomic studies, a series of slides are cut from the tissue and processed through a pipeline of steps: 1. cutting the tissue into a series of slides, 2. sample preparation, 3. RNA extraction, 4. sequencing, and 5. data analysis. Step 1-3 are usually done within a short period of time (normally several hours) to avoid sample degradation. However, with the proposed method, cutting only one slide from the tissue and processing it through the entire pipeline (steps 1-5) would take at least a week. This is because the proposed method requires waiting for the results of the first cut before deciding how to make the next cut and repeating the pipeline. This would introduce unnecessary complications and delay the progress of the study.

Author response: Thank you for this comment. We note two points in response. First, our method inherently allows for multiple slices/batches to be collected on each iteration (the experimenter is not limited to one slice). Thus, depending on time limitations, the experimenter could still use our method to inform their decisions even if they require taking larger samples on each iteration. We have adapted our manuscript and language throughout to allow a minibatch approach, where multiple slices are collected and assayed at each iteration. Second, our method is one of the first to explore the possibility of experimental design in spatial genomics and we do not aim to solve all practical issues here. We hope that our work will help lay the foundation for future work as well. We have added additional perspectives of these important points and caveats to the method in the Discussion section.

- (b) Furthermore, the proposed method is likely to compromise the quality of the samples and the data generated from the study. After the first cut, if the sample is

frozen and stored for a week, and need to be thawed before the next cut. The sample quality will degrade. This will introduce unnecessary damage to the sample and compromise the quality of the data generated from the study. Therefore, the proposed method is not only impractical but also likely to compromise the quality of the samples and the data.

Author response: Thank you for pointing this out. In discussions with experimental biologists (including collaborator Catherine Pena), the tissues are kept frozen throughout this process. In particular, mouse brains (her tissue of interest) are flash frozen after dissection and are sliced while frozen; in other words, they remain frozen throughout the (iterative) process. Thawed tissues are not of high-enough integrity to slice. If there are other samples that for some reason are thawed before slicing, we could perform a minibatch update on the first iteration, which means cutting m slices at one time selected using the same EIC function. This will remove the flexibility of selecting slices conditioned on prior slices; we could consider using alternative organs from other individuals in the process too. More importantly, our work is intended to lay the groundwork for experimental design for spatial genomics, and our intention is not to solve all practical concerns in the present work, but create a first approach that will be applicable to many but not all experimental scenarios.

2. Secondly, the manuscript uses localizing a tissue region of interest as an example to demonstrate the advantage of the proposed method in designing spatial genomics studies. However, localizing tissue regions of interest is normally done by H&E stained tissue slides, which are much faster and cost only a few dollars per slide. Thus, it is not clear why spatial genomic experiments (which cost thousands of dollars) would be used for this purpose.

Author response: Thank you for this comment. While we recognize that in a large number of cases the ROI can be identified from H&E stains alone, we focus on cases where more complex phenotyping is required to identify the ROI. We agree that a cheaper data modality should always be used if it can provide sufficient information for the task at hand. This multi-fidelity experimental design approach is a direction of our current work, in fact.

3. In summary, while the proposed method has good theoretical properties, it is not practically useful for real spatial genomic studies. Additionally, the manuscript's example of localizing tissue regions of interest using spatial genomics experiments is not convincing, given the availability of cheaper and faster alternatives.

Author response: We appreciate your thoughtful comments. We agree that there are avenues for improvement; however, we hope that this work serves

as groundwork for performing experimental design in spatial genomics studies, which has been underexplored, is currently entirely manual, and currently cannot be done outside of well-funded labs and consortia. We hope that we have convinced you that this work is an initial stepping stone to more democratic, precise, and efficient experimental design in spatial genomics.

3 Reviewer 3

3.1 Remarks to the author

Designing a cost-efficient model by slicing a tissue that can maximize information gain with fewer slices is important. This study proposed an expected information gain (EIG) model based on the expected difference in entropy between the prior and posterior to optimize the tissue slicing. They validated their model by addressing targets including region of interest localization and gene expression characterization in a tissue. The manuscript reveals an interesting methodology for spatially-resolved genomic analysis and the results are well presented. The reviewer has the following concerns:

1. For spatial transcriptomics analysis, a more critical goal is to identify spatial domains and characterize the gene expression patterns within each domain. In particular, for datasets like 10X Visium, because each spot is a mixture of multiple cells, cell type deconvolution within spots or the identification of the distributions of each cell type within tissue is also significant for ST analysis. Therefore, rather than simply characterize the gene expression patterns across the entire tissue or identify the tissue borders, the authors can test whether their model could obtain complex domain information with fewer slices.

Author response: Thank you for this suggestion. While we agree that identifying gene expression patterns is an important goal, it is not the focus of this work. Our work is specifically focused on experimental design, as the experiments can be quite costly. Methods for cell type deconvolution or gene expression pattern modeling could be applied downstream of our method after the data have been collected. We have added this as a future direction in the Discussion.

2. From the EIG algorithm, we can see that each slice is iteratively selected based on the information gained from the previous slices. As a comparison, the authors selected the “Serial” method (taking serial parallel cross sections along an anatomical plane) as a comparison. However, these parallel slices seem randomly selected in “Serial” method. It is unfair to make a comparison between the two methods. Similar to EIG model, the “Serial” method can also use the previously gained information to determine the next slice. For example, we can select the two adjacent slices which contain the largest information differences, and then use the slice in the middle of the two slices as the

next slice. This strategy is widely used in many studies. I think the authors should compare their EIG model with the new “Serial” strategy I mentioned here instead of simply randomly selecting parallel slices.

Author response: This is a great suggestion. In fact, we have already included such a method in our comparisons. The method titled *EIG (parallel)* is applying our EIG-based method but constraining the candidate slices to be parallel to one another and lie along the same plane. Given potential practical difficulties in slicing non-parallel slices, we think that the *EIG (parallel)* approach could serve as a compromise between the practicality of the *Serial* method and the efficiency of the *EIG* method.

3. In Figure 1e, Figure 3f, Figure 4b and Figure 6b, there are many cross slices within a tissue. This is not rational for a real experimental design for tissue segmentation.

Author response: Thank you for noticing this. The slices in these figures are not actually intersecting; each subsequent slice is restricted to a single fragment of tissue (and cannot intersect another slice). The full lines are drawn for clarity, but only the nonintersecting piece (within one tissue fragment) is considered as the relevant slice. We have added clarification in the captions for the relevant figures.

4. Usually, the distributions of gene expression are followed by Poisson [PMID: 33603203] or negative binomial [PMID: 35449415, 35027729, 33037292, 34686758]. However, this study used a Gaussian distribution to describe gene expression. I don’t think this assumption is rational for real spatial transcriptomics samples.

Author response: We agree that count-based likelihoods could be more appropriate for many types of spatial genomics data. Extension to count-based likelihoods is straightforward in our experimental design framework, for instance, by utilizing a Poisson Gaussian process. The variational algorithm described in our paper can be applied to this model. We agree that modeling the counts explicitly is important future work and have added an acknowledgment of this in the Discussion.

5. I don’t know how to obtain the final fragments (i.e. Figure 1g) from the highest EIG slices (i.e. Figure 1e).

Author response: Figure 1e shows candidate slices colored by the EIG associated with each of them; this panel is meant to be a pedagogical demonstration of which slices have high/low EIG. The candidate slice with the maximum EIG (the bright yellow one) is selected, and this process is repeated to obtain the slices and fragments shown in Figure 1g. We have revised the caption of Figure 1 to make this more clear.

6. “We randomly selected 100 points as observations and ran the EIG model for one iteration.” Can you explain why you need to do this step to run EIG?

Author response: This was done so that our experimental design method only had partial information (i.e., we had not observed every cell in its entirety). This was done for demonstration so that the reader could build intuition for what EIG is optimizing for (i.e., Fig. 3c shows that the border is the region with the highest information). We have added an explanation to make this more clear.

7. Suggestion: In terms of Section 4.2, the cell distributions within the three simulated tissues are all uniformly distributed. I think the authors can test their EIG model in a more extreme situation, where cells are highly dense in some locations whereas they are very sparse in some other locations.

Author response: Thank you for this comment. We have added an additional experiment that tests our approach under different distributions of cell locations. The result is in Supplementary Figure 3. We find that the performance of EIG-based design methods surpasses the Serial approach even more than in the uniform setting. This is likely due to the fact that the Serial strategy fails to optimally slice through the clusters, whereas our approach can easily slice through many cells when they are uniformly distributed.

Reviewer #1 (Remarks to the Author):

The authors addressed my previous comments and questions nicely and I believe that this paper is now ready to be published in Nature Communications.

Reviewer #2 (Remarks to the Author):

The manuscript presents an interesting approach to the design of spatial genomic studies sliding tissue samples. The proposed method has promising theoretical properties. However, my primary concern is the practical feasibility of implementing this approach in actual spatial molecular profiling experiments.

In spatial genomic studies, a well-established pipeline involves a series of sequential steps: 1) tissue sectioning into slides, 2) sample preparation, 3) RNA extraction, 4) sequencing, and 5) data analysis. Notably, steps 1 to 3 are typically executed within a brief timeframe, often several hours, to minimize sample degradation.

The proposed method deviates significantly from this conventional practice by suggesting the cutting of only one slide (or a mini-batch, as presented in the revision) from the tissue, followed by an analysis-guided decision on where to make subsequent cuts. This approach introduces several logistical challenges. Re-mounting the tissue block for each cut is prone to introducing variability into the experiment. Moreover, achieving precision in cutting at specific locations and angles, as suggested by the algorithm, proves exceedingly challenging due to the constraints imposed by the tissue block's mounting and cutting method.

Even with substantial efforts to adjust the blade's angle or the sample stand to align with the algorithm's recommended "optimum way," the resulting sections will likely exhibit uneven widths, making the remaining sections unusable. These are just a couple of logistical issues that may arise from the proposed spatial molecular profiling experiment design. It is worth noting that the only apparent benefit of this method is the potential cost savings associated with profiling fewer sections.

Given the many potential issues and challenges outlined above, it appears that the traditional serial cutting and conducting spatial profiling on all sections is a much more reliable and cost-effective approach. The conventional way of serial spatial experiments would generate significantly more information without relying on imputation and avoid the complexities introduced by the proposed experimental design.

Reviewer #3 (Remarks to the Author):

The authors have addressed my comments satisfactorily.

Response to comments on *Optimizing the design of spatial genomic studies*

Andrew Jones¹, Diana Cai², Didong Li³, and Barbara E. Engelhardt^{4,5}

¹*Department of Computer Science, Princeton University*

²*Center for Computational Mathematics, Flatiron Institute*

³*Department of Biostatistics, University of North Carolina at Chapel Hill*

⁴*Gladstone Institutes*

⁵*Department of Biomedical Data Science, Stanford University*

We thank all of the reviewers for their feedback on the paper. Below, we provide a point-by-point response to the remaining comments.

1 Reviewer 2

1.1 Remarks to the author

The manuscript presents an interesting approach to the design of spatial genomic studies sliding tissue samples. The proposed method has promising theoretical properties. However, my primary concern is the practical feasibility of implementing this approach in actual spatial molecular profiling experiments. In spatial genomic studies, a well-established pipeline involves a series of sequential steps: 1) tissue sectioning into slides, 2) sample preparation, 3) RNA extraction, 4) sequencing, and 5) data analysis. Notably, steps 1 to 3 are typically executed within a brief timeframe, often several hours, to minimize sample degradation. The proposed method deviates significantly from this conventional practice by suggesting the cutting of only one slide (or a mini-batch, as presented in the revision) from the tissue, followed by an analysis-guided decision on where to make subsequent cuts. This approach introduces several logistical challenges. Re-mounting the tissue block for each cut is prone to introducing variability into the experiment. Moreover, achieving precision in cutting at specific locations and angles, as suggested by the algorithm, proves exceedingly challenging due to the constraints imposed by the tissue block's mounting and cutting method. Even with substantial efforts to adjust the blade's angle or the sample stand to align with the algorithm's recommended "optimum way," the resulting sections will likely exhibit uneven widths, making the remaining sections unusable. These are just a couple of logistical issues that may arise from the proposed spatial molecular profiling experiment design. It is worth

noting that the only apparent benefit of this method is the potential cost savings associated with profiling fewer sections. Given the many potential issues and challenges outlined above, it appears that the traditional serial cutting and conducting spatial profiling on all sections is a much more reliable and cost-effective approach. The conventional way of serial spatial experiments would generate significantly more information without relying on imputation and avoid the complexities introduced by the proposed experimental design.

Author response: Thank you for your comments. We acknowledge the potential practical limitations brought forth by the reviewer. However, we want to emphasize a few points. As we show in the manuscript, it is possible to apply the method to many different design spaces, including those consisting of parallel cuts. As we stated in the previous revision, our goal is to lay the foundation for experimental design in spatial genomics. In particular, we envision our method being relevant to future technologies that have yet to be developed. For instance, new sequencing technologies could be developed in a budget-aware sense, given a sensible design space.